# Advanced Glycation End Products: A Sweet Flavor That Embitters Cardiovascular Disease

**DOI:** 10.3390/ijms23052404

**Published:** 2022-02-22

**Authors:** Raphael S. Pinto, Carlos A. Minanni, Aécio Lopes de Araújo Lira, Marisa Passarelli

**Affiliations:** 1Laboratório de Lípides (LIM 10), Hospital das Clínicas (HCFMUSP) da Faculdade de Medicina da Universidade de São Paulo, São Paulo 01246-000, Brazil; rspinto@usp.br (R.S.P.); carlosminanni@gmail.com (C.A.M.); aeciolira@hotmail.com (A.L.d.A.L.); 2Universidade Santa Cecília (UNISANTA), Santos 11045-907, Brazil; 3Hospital Israelita Albert Einstein (HIAE), São Paulo 05652-900, Brazil; 4Faculdade de Medicina do Centro Universitário Uninovafapi, Teresina 64073-505, Brazil; 5Programa de Pós-Graduação em Medicina, Universidade Nove de Julho (UNINOVE), São Paulo 01225-000, Brazil

**Keywords:** advanced glycation end-products, diabetes mellitus, cardiovascular disease, atherosclerosis, reverse cholesterol transport

## Abstract

Epidemiological studies demonstrate the role of early and intensive glycemic control in the prevention of micro and macrovascular disease in both type 1 and type 2 diabetes mellitus (DM). Hyperglycemia elicits several pathways related to the etiopathogenesis of cardiovascular disease (CVD), including the generation of advanced glycation end products (AGEs). In this review, we revisit the role played by AGEs in CVD based in clinical trials and experimental evidence. Mechanistic aspects concerning the recognition of AGEs by the advanced glycosylation end product-specific receptor (AGER) and its counterpart, the dolichyl-diphosphooligosaccharide-protein glycosyltransferase (DDOST) and soluble AGER are discussed. A special focus is offered to the AGE-elicited pathways that promote cholesterol accumulation in the arterial wall by enhanced oxidative stress, inflammation, endoplasmic reticulum stress and impairment in the reverse cholesterol transport (RCT).

## 1. Introduction

The last two decades have been marked by alarming data about the epidemic of diabetes mellitus (DM). In 2021, an estimated 537 million adults aged 20–79 years have DM. This represents a prevalence of 10.5% of the world’s population in this age group. This number is projected to reach 643 million by 2030, and 783 million by 2045, according to the International Diabetes Federation (IDF) [1].

DM confers a higher risk of development of chronic complications and people with DM are 2 to 3 times more likely to have cardiovascular disease (CVD) [2,3]; the prevalence of end-stage renal disease is 10 times higher [4] and amputation is 10 to 20 times more common [5] in people with DM as compared to the healthy population. In addition, diabetic retinopathy is the leading cause of vision loss in working-age adults [6].

The classic studies, namely the United Kingdom Prospective Diabetes Study (UKPDS), Action in Diabetes and Vascular Disease: Preterax and Diamicron Modified Release Controlled Evaluation (ADVANCE), Action to Control Cardiovascular Risk in Diabetes (ACCORD), and Diabetes Control and Complications Trial/Epidemiology of Diabetes Interventions and Complications (DCCT/EDIC), demonstrated that intensive glycemic therapy in patients with a recent diagnosis of type 1 and 2 diabetes (T2DM) was associated with a reduced risk in long-term complications, in particular, microvascular morbidities. Data from 30 years of the DCCT/EDIC demonstrated that intensive versus conventional glycemic treatment also has beneficial effects on macrovascular disease, especially on subclinical markers of atherosclerosis (carotid intima-media thickness and coronary calcium score) 6–12 years after the end of randomized treatment. Intensive treatment reduced aggregate cardiovascular risk by 42% and major cardiovascular events by 57% (myocardial infarction, stroke, and cardiovascular death) [7]. However, in many cases, when the glycemic control was achieved after a period of intense decompensation, the development of chronic complications was not well prevented [8,9]. Those findings seem to rely on the concept of metabolic memory or legacy effect that is based on both the permanent modification of biological macromolecules during the period of glycemic decompensation and on the accumulated risk memorized by long-term exposure to oxidative stress and advanced glycation products (AGEs), which may favor epigenetic changes. The risk of complications remains even with multifactorial interventions based on glycemic-reducing agents and antilipemic and antihypertensive drugs [8,10].

Hyperglycemia is the major etiopathogenic factor for the development of DM complications that is due to the induction of oxidative stress, considered the basis for cellular and tissue damage. Hyperglycemia favors the generation of AGEs, induces the activation of hexosamine, polyol, and protein kinase C pathways that exacerbate oxidative stress. In this review, we summarize the data on the role played by AGEs in the pathophysiology of cardiovascular disease in DM, based on human clinical studies and experimental models.

## 2. Formation of Advanced Glycation End Products

Louis Camille Maillard first described in 1912 the nonenzymatic reaction between reducing sugars with foods in what was called a browning reaction. Patton and Hill (1948) [11] characterized the chemical reaction that attained much more attention by the identification by Rahbar et al. (1969) [12] of a glucose-modified form of hemoglobin (glycated hemoglobin) that was found elevated in the blood of subjects with DM as compared to healthy controls. Since then, HbA1c was utilized as a parameter of metabolic control and management of DM and was associated with the development of DM complications.

The covalent and nonenzymatic reaction between reducing sugars, including glucose, with the amino-terminal portion of lysine and arginine residues in proteins, phospholipids, and nucleic acids leads to the formation of an unstable Schiff Base. The Schiff Base undergoes a more stable Amadori product or fructosyl lysine (such as glycated hemoglobin and fructosamine) and this reaction primarily depends on plasma glucose levels and the time of DM decompensation. Molecular intra and inter rearrangements of the Amadori product in most of the cases involve oxidative reactions leading to the generation of oxoaldehyde that induces a rapid and irreversible modification of proteins by advanced glycation. Oxoaldehydes are very reactive compounds, such as glyoxal (GO), methylglyoxal (MGO), glycolaldehyde, and 3-deoxyglucosone (3-DG), which are generated in DM, but also in other metabolic conditions where the carbonyl stress prevails. They rapidly interact with macromolecules leading to the fast formation of extracellular and intracellular AGEs (Figure 1).

Intracellular generated AGEs originate from the increased glucose flow through glycolysis that favor the output of reactive oxygen species (ROS) from mitochondria and the DNA-repairing enzyme poly (ADP-ribose) polymerase (PARP). Although protecting DNA from ROS-induced damage, PARP promotes poly ribosylation of the glyceraldehyde 3-phosphate dehydrogenase (GAPDH), impairing its activity. Substrate deviation from glycolysis generates MGO and, as a consequence, AGEs are formed feeding a vicious circle by inducing oxidative stress [13,14]. Hyperglycemia also induces the activation of the hexosamine, polyol and protein kinase C pathways, exacerbating the intracellular oxidative stress, which perpetuates the AGE generation [13,15] (Figure 2).

Independently of hyperglycemia, AGEs are induced during acute and chronic inflammation due to the activation of the neutrophil myeloperoxidase that induces glycolaldehyde (GAD), another very reactive oxoaldehyde. Additionally, the oxidation of certain amino acids, polyunsaturated fatty acids, and ketone bodies contributes to the generation of oxoaldehydes and then to AGE generation. In chronic kidney disease, AGEs are formed due to a reduction in the clearance of glycation reaction precursors and reduced antioxidant enzymes. Then, although the glycation reaction was firstly ascribed to a hyperglycemic milieu, it is now conceivable that it also occurs independently of glucose concentration in plasma and tissues [16]. In fact, glycation is prevalent in healthy aging individuals, mainly in association with long half-life protein, such as collagen and crystalline. Although this process takes place continuously in the body during aging, it is extremely accelerated in DM and in other carbonyl stress conditions.

Exogenous sources of oxoaldehydes and AGEs include diet and tobacco, representing unpredictable sources of the AGEs that contribute the body’s AGE pool, although the exact mechanisms that regulate AGE absorption are not well known. The impact of AGE-containing diets on human health is under intensive investigation. High heating, dry cooking, and long-time processing increase AGEs in baked, broiled, grilled, and fried foods. Cooking in an acidic medium containing wine, vinegar, or citrus juice helps to prevent AGE formation in food. A recent publication demonstrated an endocytic pathway via scavenger receptors mediating the intestinal absorption of dietary carboxymethyllysine in *C. elegans* [17]. In healthy humans, about 10% of dietary AGEs are absorbed and transported in circulation in association with albumin and lipoproteins, and 30% is eliminated in the urine. On the other hand, in individuals with kidney disease, only 5% of dietary AGE is excreted in the urine [18]. A high-AGE diet results in significant elevations in serum AGE and induces oxidative stress [19]. Oxoaldehydes increase in the postprandial states and during glycemic excursions inducing a fast generation of AGEs in circulation [20]. Glucose fluctuations increase plasma levels of glyceraldehyde-derived advanced glycation end-products and relate to the severity of cardiovascular disease in DM [21].

Advanced glycation end products are very heterogeneous, making difficult their assessment in blood, tissues, and cells. Some of them induces protein crosslinking and/or emit fluorescence [22]. Glyoxal and GAD render carboxymethyl lysine (CML) the most abundant AGE found in the body [23,24], particularly in the target tissues of DM late complications, such as kidneys, eyes, skin, and vessels [25,26,27,28,29]. Methylglyoxal renders to carboxyethyl lysine, MGO-derived hydroimidazolone 1 (MGO-H1), and MGO-lysine dimer, although other structures are also described in plasma and tissues, such as pentosidine, pirralyne, argypirimidine, and GO- lysine dimer.

AGEs are recognized by several receptors, although the signaling pathway elicited is better described for the advanced glycosylation end product-specific receptor, (alias RAGE). In fact, it is demonstrated that AGER mediates the biological effects of AGEs in many cell types. AGER is a multiligand receptor of the immunoglobulin superfamily, a member of pattern-recognition receptors that recognize AGEs, but also calgranulins, high mobility group protein B1 (HMGB1), lipopolysaccharides (LPS), sheet fibrils, and phosphatidylserine on the surface of apoptotic cells. The receptor for AGEs actively participates in diabetic vascular complications, as well as in the interface of innate and adaptive immunity and in inflammation. In this sense, there is a lot of evidence supporting the concept that AGEs and AGERs play an active role in the development and progression of cardiovascular disease in DM [30].

A sequence of 394 amino acids in a single hydrophobic transmembrane domain (19 amino acids) and a highly charged C-terminal cytosolic tail (43 amino acids) that mediates intracellular signaling pathways characterizes the AGER structure. The extracellular portion contains an N-terminal immunoglobulin (Ig) V-type ligand-binding domain and two Ig C-type domains (V-C-C′). The protein diaphanous homolog 1 (DIAPH1) mediates the activation of NADPH oxidase 4 after AGE binding to the AGER. The canonical pathway involves ROS generation and the transactivation of the AGER gene and others related in the production of inflammatory cytokines, adhesion molecules, chemokines, growth factors, and scavenger receptors [31,32,33]. Other signaling pathways stimulated by the AGE–AGER interaction include extracellular signal-regulated kinase 1/2 (ERK 1/2) and p38-mitogen-activated protein kinase (MAPK), PI3K/AKT, Rho GTPases, and cross talking with Toll-like receptors [34,35]. Interestingly, it is observed that the expression of the AGER is low in vascular cells, but constitutively activated in DM and inflammation [36].

Soluble isoforms of AGERs lacking intracellular domains bind to AGEs, but are unable to trigger cell signaling. There are two isoforms of soluble AGER, although their regulation is not well known. The first isoform is produced by the action proteases at cell surface, such as disintegrin and metalloproteinase domain-containing protein 10, (ADAM10) and matrix metalloproteinases (MMPs). The second isoform called endogenous secretory, esAGER, is generated by alternative splicing of the AGER gene. The measurement of a soluble AGER (sAGER) in circulation was proposed as a biomarker of CVD in DM. However, data in the literature are still controversial.

The dolichyl-diphosphooligosaccharide-protein glycosyltransferase (DDOST, alias AGER1) binds AGEs, although the intracellular signaling elicited by their interaction is not well known. It is encoded by the DDOST gene and is expressed in different cell types, including macrophages [36,37,38]. In humans, the concentration of DDOST correlates inversely with intracellular concentrations of AGEs and directly with urinary AGEs [39]. The dolichyl-diphosphooligosaccharide-protein glycosyltransferase mediates the uptake and degradation of AGEs and inhibits the activity of NADPH oxidase, which prevents the activation of NF-KB [36,40]. In addition, DDOST also prevents the epidermal growth factor receptor activation via AGEs, in the presence of the oxidative insult, and may play an important role in restricting the activity of other G protein-coupled receptors [36,37]. In mice, high concentrations of DDOST prevented the formation of atheroma induced by a high-fat diet or by DM [41]. It is still unknown if the determination of the AGER/DDOST ratio can discriminate the susceptibility of different tissues to hyperglycemia-induced complications.

Many other AGE receptors and soluble binding proteins interacting with AGEs are described, including macrophage scavenger receptors types I and II (MSR-I), class B member 1 (SRB1), galectin-3-binding protein, FEEL-1 and 2 (stabilin-1 and stabilin-2) [42], and Toll-like receptors [35].

The degradation of AGEs seems to involve lysozyme and lactoferrin-like polypeptide, while amadoriases and other enzymes mediate the degradation of intermediates of the glycation reaction, although the role of those proteins in the AGE homeostasis is not fully understood. The enzymes glyoxalase (GLO) 1 and 2 play key roles, by detoxifying MGO into SD-lactoylgluthatione and D-lactate, respectively, reducing the AGE burden [43,44].

## 3. AGEs as Indicators of Cardiovascular Burden: Where Are We?

The body pool of AGEs is determined by carbonyl and inflammatory stress together with kidney function that drives endogenous AGEs formation. In addition, exogenous sources that are in most cases unpredictable and variable add to this pool worsening chronic complications elicited by the AGE accumulation (Figure 3).

Markers of early glycation, such as HbA1c, fructosamine, and glycated albumin, were associated with vascular outcomes and mortality in the community-based Atherosclerosis Risk in Communities (ARIC) study [45]. Additionally, AGEs independently relate to atherosclerosis in both DM and non-DM subjects [46,47,48,49]. Strong evidence demonstrates that in DM patients [50] or DM patients with micro and/or macrovascular complications [51,52] and individuals with chronic kidney disease [53], circulating AGEs are elevated and associated with the progression of complications. Particularly, CML levels predict arterial stiffness [30] and the severity of vascular obstruction [54].

In the CORDIOPREV study, higher levels of MGO were found in T2DM subjects with severe endothelial dysfunction and increased intima-media thickness that are subclinical markers of atherosclerosis. Independently of the endothelial dysfunction, plasma CML levels were increased in people with established T2DM as compared to newly diagnosed DM individuals [55].

In a 64-year-old man with poorly controlled T2DM, obesity, smoking, hypertension, and dyslipidemia over 15 years, the absence of DM-related complications was related to low levels of AGEs (CEL and MG-H1) and sAGERs in plasma. These findings suggest that the AGE/RAGE axis may sensitize to DM-related complications [56].

The Japan Assessment of Pitavastatin and Atorvastatin in Acute Coronary Syndrome (JAPAN-ACS) did not show a correlation between the AGE and soluble AGER (sAGER) levels with atherosclerotic plaque volume. Despite the baseline levels of the AGE and sAGER were similar between the DM and non-DM subjects, the higher AGE levels were associated with plaque progression [57].

Although sAGERs seem to play a protective role in neutralizing the toxic action of AGEs, there are divergent data in the literature regarding its role in CV disease. Those conflicting results may be related to the possible distinct role of the two different isoforms of sAGERs.

In individuals diagnosed with pre-diabetes (HbA1c 5.7–6.4%) subclinical cardiac alterations related to the left atrium volume, the atrial filling velocity was independently associated with the soluble sAGER [58]. In addition, sAGERs and calgranulin S-100B were considered important biomarkers for early hemorrhagic and ischemic stroke differentiation [59]. Soluble AGER was increased in patients who underwent coronary artery bypass graft with the worst outcome as compared to those with better clinical results [60].

In normal glucose tolerance subjects, one-hour post-load glycemia was associated with low sAGERs and with increased S100A12 calgranulin levels, pulse wave velocity, and intima-media thickness [61]. Interestingly, the offspring of patients with early onset of CVD presented lower levels of sAGERs in comparison to age-matched healthy controls, suggesting that the sAGER measurement might be a valuable predictor for the stratification of CV risk [62].

A decrease in the sAGER concentration was demonstrated in T2DM subjects and associated with increased oxidative stress and endothelial dysfunction in DM [63,64]. In T1DM subjects, decreased sAGERs were found and was inversely associated with the progression of the atherosclerotic lesion. However, a positive correlation between sAGERs and CVD mortality was demonstrated in these individuals [65].

In T2DM, the ADVANCE study demonstrated a positive correlation between sAGERs and circulating AGEs, being both associated with genesis and progression of nephropathy. Additionally, sAGERs were correlated with all causes mortality [66].

The serum levels of glycated albumin and sAGERs (endogenous secretory AGERs) were, respectively, positive and negatively associated with coronary artery remodeling in DM 2 [67]. In addition, the sAGER independently correlates with the atherosclerotic lesion in patients with hypertension [68] and is an independent predictor of the worst prognosis in heart failure [69]. Contrastingly, Nakamura et al. (2007) [70] demonstrated an increase in sAGERs in T2DM patients when compared to non-DM subjects, and this correlated with the presence of CVD. Corroborating these findings, in a prospective study it was found that in T2DM individuals the high sAGER concentration was a predictor of CV events [71]. In a 12-year follow-up of a general population in Germany, AGEs and sAGERs were not associated with all-cause mortality in both men and women [72].

Reichert et al. followed 886 patients with CVD (acute myocardial infarction, stroke/transient ischemic attack-TIA) for 3 years, and observed that the presence of sAGERs in the upper quartile correlated with an increased incidence of CVD recurrence (24.9% vs. 13.1%, *p* < 0.0001), which was confirmed even after multivariate regression for possible confounders [73]. Basta et al. (2009) showed in a symptomatic patient who underwent carotid endarterectomy elevated sAGER concentrations in plasma as compared to asymptomatic subjects, suggesting that the sAGER may be an indicator of a higher degree of vascular inflammation [74]. Recently, our group demonstrated an inverse association between the concentrations of plasma sAGERs with oxysterols in advanced carotid atherosclerotic lesions. Furthermore, plasma fluorescent AGEs were positively related to oxysterol content and markers of cholesterol synthesis and absorption. These data indicate that sAGERs and plasma AGEs can be used as a tool to infer sterol accumulation in arteries, helping to prevent and manage acute vascular complications [70].

Despite the controversies, sAGERs are investigated as a therapeutic tool to counteract the deleterious effects of AGEs in experimental models. In addition, Ager silencing or Ager knockout animal models are protected from DM complications [75,76]. In diabetic Apoe knockout mice, treatment with sAGERs decreased atherosclerotic lesion, regardless of plasma glucose and lipid concentration, suggesting that AGER may be an important therapeutic target for diabetic macrovascular disease [77,78]. In atherosclerotic plaques, even in the absence of DM, there is a higher content of AGER ligands, such as AGEs, HMGB1, and S100/calgranulins, strengthening the idea of participation in the AGER signaling in the etiopathogenesis of atherosclerosis [79,80].

Skin tissue AGEs, assessed by SAF, are associated with the development of macrovascular events after a 5-year follow-up in T2DM [81] and are considered as good indicators of atherosclerotic burden [82]. In addition, independently of DM and CV risk factors, SAF was associated with carotid atherosclerosis in the elderly [83]. Despite not all being fluorescents, tissue AGEs are well correlated with skin autofluorescence (SAF). In skin-collagen biopsies from T1DM subjects enrolled in the DCCT/EDIC study, pentosidine, MG-H1, and glucosepane were found in association with the intima-media thickness (IMT) [84].

## 4. Sweet AGEs Driving the Bitter Pathophysiology of CVD

Although firstly described in association with long half-life proteins, such as crystalline and collagen, advanced glycation also occurs in long and short half-life proteins, creating a large spectrum of biological derangements. Apolipoproteins and some phospholipids that are constituents of lipoproteins are susceptible to advanced glycation compromising their biological functions. Chylomicrons and very-low-density lipoproteins (VLDL) are less susceptible to the lipoprotein lipase-mediated hydrolysis of triglycerides favoring hypertriglyceridemia. Due to the enhanced affinity of cholesteryl ester transfer protein for glycated lipoproteins and due to hypertriglyceridemia, an enhanced exchange of esterified cholesterol and triglycerides can be observed between TG-rich lipoproteins and low (LDLs) and high-density lipoproteins (HDLs). This favors the formation of small dense LDLs that are atherogenic, being more susceptible to oxidation and to enter the arterial wall compartment. Glycated LDLs are not well recognized by the LDL receptor (B-E receptor), allowing this particle to reach the arterial wall, where they are taken up by macrophage receptors (AGERs, Toll-like receptors, and scavenger receptors class A and B) leading to intracellular cholesterol accumulation and foam cell formation.

Contrary to LDLs, which have a longer half-life when glycated, glucose-modified HDLs are faster removed from the plasma. Considering the role of this lipoprotein in the prevention of atherosclerosis, there are several investigators dealing with HDL glycation and the consequences to its function.

An HDL mediates the reverse cholesterol transport a unique system that promotes excess cholesterol exportation from arterial wall macrophages to the liver, allowing its excretion in feces. Lipid poor-apoA-I or nascent HDL particles (pre-beta HDLs) interact with the phospholipid-transporting ATPase ABCA1 (ABCA-1) that is expressed in macrophages by the liver X receptor (LXR) activation by oxysterols. Free cholesterol is esterified by the phosphatidylcholine-sterol acyltransferase (alias, lecithin cholesterol acyltransferase; LCAT) being transferred to the hydrophobic HDL core that assumes a spherical format. Larger HDL particles, especially HDL2, remove more cell cholesterol by the interaction with the ATP binding cassette transporter G-1 (ABGA-1) that also exports toxic oxysterols outside macrophages. Esterified cholesterol is transferred to apo B-containing lipoproteins (chylomicrons, VLDL and LDL) by the action of CETP; allowing for the uptake of these lipoproteins by the hepatic receptors B-E and the prolow-density lipoprotein receptor-related protein 1 (LRP-1). The scavenger receptor class B member 1 (SR-B1) in the liver selectively removes esterified cholesterol in HDLs. Cholesterol can be eliminated in bile in its free form or after conversion into cholic and deoxycholic acids, by the action of, respectively, oxysterol 7-alpha-hydroxylase and sterol 27-hydroxylase, and then excreted in feces.

Early and advanced glycation compromises many steps of the RCT contributing to atherogenesis in DM. The LCAT activity on glycated HDLs is reduced, while the CETP activity is enhanced favoring the accumulation of cholesterol in atherogenic lipoproteins (LDLs, VLDLs, and chylomicrons) [85]. In addition, advanced glycated HDLs remove less cholesterol from macrophages, and the esterified cholesterol delivery to the liver mediated by SRB1 is reduced [86] (Figure 4).

Albumin is the major serum protein modified by advanced glycation and it is clinically utilized to infer glycemic control. Our group focused on analyzing the role of advanced glycated albumin (AGE-albumin) in lipid macrophage homeostasis. Albumin modified by AGEs is taken up by macrophages via AGERs and induces ROS generation by the NADPH oxidase 4 and the mitochondria [87]. In addition, the AGE–AGER elicits the production of inflammatory cytokines that together with ROS and the accumulation of toxic oxysterols (mainly 7-ketocholesterol) induce endoplasmic reticulum (ER) stress [88,89,90]. The ER stress is linked to proteasomal degradation of proteins that induces the degradation of ABCA-1 in macrophages incubated with AGE-albumin produced in vitro by the incubation with oxoaldehydes or isolated from DM individuals serum [91,92,93]. Ultimately, these events lead to the intracellular accumulation of cholesterol and oxysterols contributing to atherogenesis (Figure 4).

The intracellular accumulation of 7-ketocholesterol and the ER stress are related to plaque instability and rupture [94] due to cell necrosis and apoptosis [95]. Additionally, in vivo glycated albumin induces extracellular matrix derangement and the expression of adhesion molecules related to vascular damage [96]. All these data reinforce the clinical observation that AGE-albumin in serum is a marker for DM complications, especially CV disease.

Macrophages incubated with albumin isolated from poorly controlled DM subjects showed reduced cholesterol efflux, increased the secretion of inflammatory cytokines induced by LPS, and increased the ABCA-1 degradation rate [92,93]. Those events were normalized when those cells were incubated with albumin from the same subjects after improvement of the glycemic control [97]. Moreover, the effects of AGE-albumin in impairing cholesterol removal and inducing inflammation were sustained in cells even after the removal of AGE-albumin from the incubating medium, reflecting the long-lasting effects of advanced glycated albumin on macrophage lipid homeostasis derangements and inflammation [85]. Recent studies [98,99] demonstrate that albumin and HDLs isolated from patients with diabetes and chronic kidney disease, with decreased glomerular filtration, diminishes the efflux of cholesterol mediated by HDLs in macrophages, which may be a result of glycation and the carbamoylation that occurs in these patients with diabetes and CKD. Furthermore, hyperglycemia, by inducing ROS, promotes epigenetic changes that favor NF-KB activation and cholesterol efflux down-regulation [100].

Interestingly, AGER silencing as well the in vitro treatment of macrophages with AGE-blockers and antioxidants were able to prevent disturbances in the RCT [101,102,103]. In other studies, AGER knockdown or treatment with sAGERs were able to prevent atherogenesis in dyslipidemic animal models [77,104].

In a non-DM dyslipidemic animal model (Apoe knockout mouse), it was observed that AGE-albumin chronically injected in mice peritoneum lead to accelerated lipid deposition in the aortic arch as compared to non-glycated albumin. The amount of CML, AGER, and 4-hydroxynonenal, a marker of lipid peroxidation, greatly increased in the arterial wall of AGE-albumin-treated animals. Then, even in the absence of DM, AGEs by themselves triggered atherogenesis in dyslipidemic mice [105].

Finally, it is noteworthy that in addition to being greatly induced by hyperglycemia and inflammation that prevail in DM, AGEs by themselves can induce DM by diminishing peripheral insulin sensitivity and beta-cell insulin secretion [106]. We recently demonstrated that in healthy rats, AGEs that were chronically injected increased body weight and induced whole-body insulin resistance by reducing the expression of the Scl2a4 gene and its product, glucose transporter 4 (Glut4), in peri-epidydimal adipocytes [107] and skeletal muscle. In the soleus muscle, an increased amount of NFKB (p50) in the cell nucleus and the elevated cell content of the ER stress/unfolded response marker [106] were observed. These findings are particularly relevant when considering that Western-type diets are enriched in AGEs and can impact human health [108,109].

## 5. AGEs: Where Are We Going?

Advanced glycation end products (AGEs) have emerged as promising new biomarkers for the development of CVD in DM and other chronic conditions, such as chronic kidney disease (CKD), inflammatory diseases, and obesity [30,110,111]. This is based on their pathophysiological involvement and/or prediction of lipid and glucose metabolism derangements, as well as oxidative and inflammatory stress that aggravate cholesterol accumulation in the arterial wall.

However, there are some pitfalls regarding its use in clinical practice that must be circumvented. The complexity of the high heterogeneous AGE structure and their rapid interconversion make it difficult to access their specific role in CV disease. Antibody-based AGE detection, including enzyme-linked immunosorbent assay and immunoblot, are used to measure specific AGEs in plasma, tissues, and cell culture lysates, although those techniques are quite variable due to a great variety of epitopes that can be detected. On the other hand, liquid chromatography/mass spectrometry analysis can provide a more reliable profile of different AGE structures, despite being a costly and laborious method, making it difficult to analyze a large number of samples. A great variety of AGEs can be noninvasively detected by skin autofluorescence (SAF) [112]. SAF correlates with plasma and tissue AGEs, although confounders, such as ethnicity, age, self-browning creams, and skin adiposity [113], may oppose its use in the clinical prediction of DM complications.

The nature of the cross-sectional design of the majority of studies has limited conclusive elucidations regarding the role of AGEs and soluble AGERs in the development of complications associated with chronic diseases. Finally, it is difficult to address the impact of high-AGE-containing foods on health, considering the ethical limitations for long-term studies, the intrinsic presence of high fat and other nutrients fairly related to CV risk, and the presence of comorbidities.

## 6. Conclusions

Many clinical studies have pointed to AGEs as independent risk markers for CVD, which have been supported by studies in cell culture and animal models. All those data have added to our growing knowledge of AGE structures, measurement, stability, and biological actions opening new routes for future therapies based on the AGE inhibition and AGE–AGER axis blocking. Many methodological techniques have been intended to establish AGEs as a biomarker for CVD, which should be proven in larger longitudinal designed studies.

## Figures and Tables

**Figure 1 ijms-23-02404-f001:**
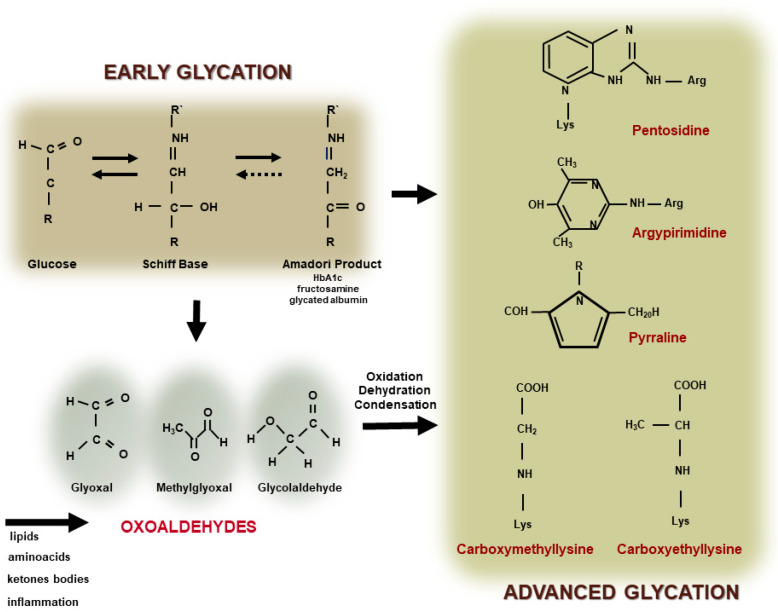
Advanced glycation end product (AGE) formation. The glycation reaction takes place by the modification of amino-terminal groups of proteins, phospholipids, or nucleic acids, by glucose or other monosaccharides leading to the generation of a Schiff Base and an Amadori product (early glycation). The auto-oxidation of glucose and rearrangements of the Schiff Base and the Amadori product, as well as the oxidation of amino acids, lipids or ketone bodies, and inflammation, promotes the generation of oxoaldehydes (such as glyoxal, methylglyoxal, and glycolaldehyde). They lead to the rapid and irreversible generation of AGEs, including heterogenous compounds, such as, pentosidine, argypirimidine, pyrraline, carboxymethyllysine, and carboxyethyllysine.

**Figure 2 ijms-23-02404-f002:**
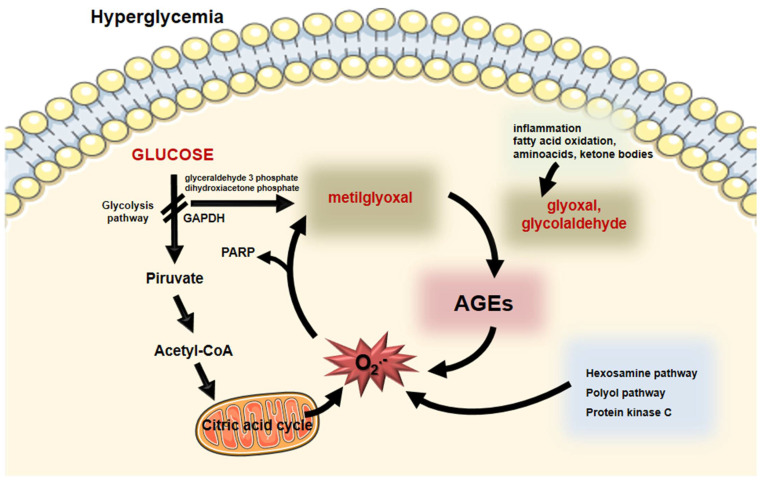
Intracellular formation of AGEs. During hyperglycemia, enhanced glucose flux through glycolysis induces the generation of superoxide anion by the mitochondria. Poly (ADP- ribose) polymerase (PARP) that protects DNA from cleavage induces a posttranslational modification of the glyceraldehyde 3P-dehydrogenase (GADPH) impairing glycolysis. Substrate (glyceraldehyde 3 phosphate and dihydroxyacetone phosphate) deviation leads to the generation of methylglyoxal that induces the formation of AGEs. Moreover, inflammation, fatty acid, amino acid, and ketone body oxidation generate oxoaldehydes, including glyoxal and glycolaldehyde that promote AGE generation. Oxidative stress increases by AGEs and by other biochemical pathways elicited by hyperglycemia (hexosamine, polyol, and protein kinase C). This vicious circle feeds the formation of AGEs and oxidative stress that is the base of cellular complications in diabetes mellitus. Parts of the figure were drawn using Servier Medical Art (https://smart.servier.com/, accessed on 14 December 2021).

**Figure 3 ijms-23-02404-f003:**
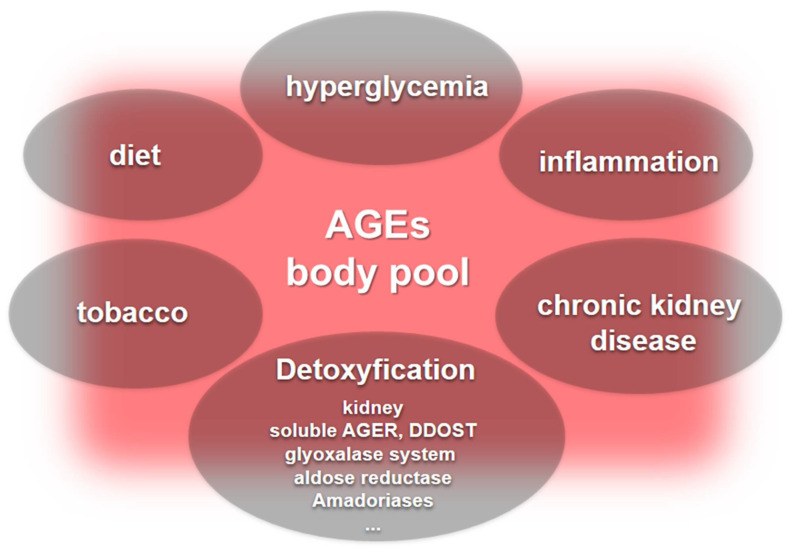
The body pool of AGEs. Hyperglycemia, inflammation, and chronic kidney disease are major contributors to the body’s pool of AGEs. Furthermore, dietary sources of AGEs and tobacco add to the AGE pool, while detoxification systems include kidney function and the role played by soluble AGERs and DDOST that neutralize deleterious AGER signaling, along with enzyme systems, including the glyoxalase system, aldose reductase, and the lesser-known amadoriases.

**Figure 4 ijms-23-02404-f004:**
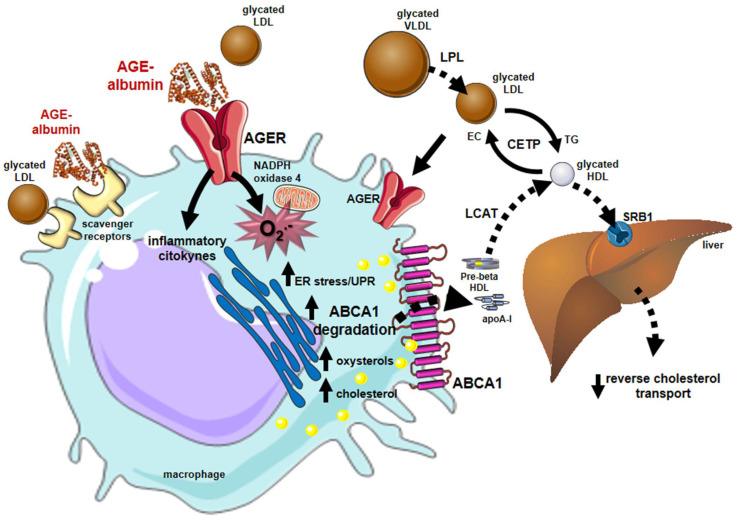
Major influences of glycation on lipid metabolism and reverse cholesterol transport that favor atherogenesis. Alterations in lipid and lipoprotein metabolism induced by early and advanced glycation favor atherogenesis. In blood circulation, the lipolysis of chylomicrons and VLDLs by the lipoprotein lipase (LPL) is damaged by glycation, inducing plasma triglycerides elevation. The activity of CETP is also stimulated between glycated lipoproteins enabling more cholesterol to be transferred from HDLs to atherogenic lipoproteins and reducing HDL cholesterol. Glycated LDLs are taken up by macrophage scavenger receptors and by the advanced glycosylation end product-specific receptor (AGER). Advanced glycated albumin (AGE-albumin) induces, via AGERs, oxidative stress by enhancing the activity of NADPH oxidase 4 and mitochondrial systems and inflammation. Together, oxidative stress and inflammation lead to ER stress that triggers proteasomal and lysosomal degradation of the ABCA-1 compromising the cholesterol efflux to apo A-I and the generation of HDLs. The activity of (LCAT) is impaired by HDL glycation reducing the lipoprotein maturation. The accumulation of cholesterol and toxic oxysterols in macrophages triggers oxidation, inflammation, and ER stress creating a vicious circle that contributes to atherogenesis, plaque instability, and rupture. Parts of the figure were drawn using Servier Medical Art (https://smart.servier.com/, accessed on 14 December 2021).

## Data Availability

Not applicable.

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
