# Peer review of "Advanced Glycation End Products: A Sweet Flavor That Embitters Cardiovascular Disease"

_ijms, 2022, doi:10.3390/ijms23052404_

Round 1

Reviewer 1 Report

The review is very interesting and creates an efficient point of view about the role that AGEs play in CVD diseases. The figures included are clear and support the readers to better understand the biological mechanisms that AGEs play.

Author Response

The authors would like to thank the reviewer for his comments.

Reviewer 2 Report

I think the authors propose an appropriate and very important topic.

The manuscript is well organized and reads quite easily, even the figures are clear and well organized.

I would suggest also consider the impact of AGEs coming from food (in particular due to cooking) as they increase the global content, I would better characterize the quote related to figure 3

Also, the nutritional aspect in particular with regard to glycemic peaks should at least be considered.

Author Response

The authors would like to thank the reviewer for suggestions that helped to improve the manuscript. The following sebtebce was included in the new version of the manuscript.

“The impact of AGE-containing diets on human health is under intensive investigation. High heating, dry cooking, and long-time processing increase AGE in baked, broiled, grilled, and fried foods. Cooking in an acidic medium containing wine, vinegar, or citrus juice helps to prevent AGE formation in food. A recent publication demonstrated an endocytic pathway via scavenger receptors mediating the intestinal absorption of dietary carboxymethyllysine in C. elegans [17]. In healthy humans, about 10% of dietary AGEs are absorved and transported in circulation in association with albumin and lipoproteins, and 30% is eliminated in the urine. On the other hand, in individuals with kidney disease, only 5% of dietary-AGE is excreted in the urine [18]. A high-AGE diet results in significante elevations in serum AGE post-feeding and induces oxidative stress [19]. Oxoaldehydes increase in the postprandial states and during glycemic excursions inducind a fast generation of AGEs in circulation [20]. Glucose flutctuations increase plasma levels of glyceraldehyde-derived advanced glycation end-products and relate to the severity of cardiovascular disease [21].“